# LDMol: A Text-to-Molecule Diffusion Model with Structurally Informative Latent Space Surpasses AR Models

**Jinho Chang**[1]   **Jong Chul Ye**[1]

## Abstract

With the emergence of diffusion models as a frontline generative model, many researchers have proposed molecule generation techniques with conditional diffusion models. However, the unavoidable discreteness of a molecule makes it difficult for a diffusion model to connect raw data with highly complex conditions like natural language. To address this, here we present a novel latent diffusion model dubbed LDMol for text-conditioned molecule generation. By recognizing that the suitable latent space design is the key to the diffusion model performance, we employ a contrastive learning strategy to extract novel feature space from text data that embeds the unique characteristics of the molecule structure. Experiments show that LDMol outperforms the existing autoregressive baselines on the text-to-molecule generation benchmark, being one of the first diffusion models that outperforms autoregressive models in textual data generation with a better choice of the latent domain. Furthermore, we show that LDMol can be applied to downstream tasks such as molecule-to-text retrieval and text-guided molecule editing, demonstrating its versatility as a diffusion model.

## 1. Introduction

Designing compounds with the desired characteristics is the essence of solving many chemical tasks. Inspired by the rapid development of generative models in the last decades, *de novo* molecule generation via deep learning models has been extensively studied. Diverse models have been proposed for generating molecules that agree with a given condition on various data modalities, including string representations (Segler et al., 2017), molecular

graphs (Lim et al., 2020), and point clouds (Hoogeboom et al., 2022). The attributes controlled by these models evolved from simple chemical properties (Olivecrona et al., 2017; Gómez-Bombarelli et al., 2018) to complex biological activity (Staszak et al., 2022) and multi-objective conditioning (Li et al., 2018; Chang & Ye, 2024). More recently, as deep learning models' natural language comprehension ability has rapidly increased, there's a growing interest in molecule generation controlled by natural language (Edwards et al., 2022; Pei et al., 2023; Liu et al., 2024a; Su et al., 2022) which encompasses much broader and user-friendly controllable conditions.

Meanwhile, diffusion models (Song & Ermon, 2019; Ho et al., 2020) have emerged as a frontline of generative models over the past few years. Through a simple and stable training objective of predicting noise from noisy data (Ho et al., 2020), diffusion models have achieved highly realistic and controllable data generation (Dhariwal & Nichol, 2021; Karras et al., 2022; Ho & Salimans, 2021). Furthermore, leveraging that the score function of the data distribution is learned in their training (Song et al., 2021b), state-of-the-art image diffusion models enabled various applications on the image domain (Saharia et al., 2022; Kim & Ye, 2021; Chung et al., 2023). Inspired by the success of diffusion models, several papers suggested diffusion-based molecule generative models on various molecule domains including a molecular graph (Luo et al., 2023), strings like Simplified Molecular-Input Line-Entry System (SMILES) (Gong et al., 2024), and point clouds (Hoogeboom et al., 2022).

However, a discrepancy between molecule data and common data domains like images makes it hard to connect the diffusion models to molecule generation. Whereas diffusion models are deeply studied on a continuous data domain with Gaussian noise, any molecule modality has inevitable discreteness such as atom and bond type, connectivity, and SMILES tokens (Figure 1-(a)). As a result, diffusion models trained on raw molecule data often failed to faithfully follow the given conditions or showed poor data quality (*e.g.*, invalid molecules) as the condition became more sophisticated like natural language. Most molecule diffusion models presented so far have used a few, relatively simple conditions to control, while major developments in text-to-molecule

---

[1]Graduate School of Artificial Intelligence, Korea Advanced Institute of Science and Technology, Seoul, South Korea. Correspondence to: Jong Chul Ye <jong.ye@kaist.ac.kr>.

*Proceedings of the 42nd International Conference on Machine Learning*, Vancouver, Canada. PMLR 267, 2025. Copyright 2025 by the author(s).

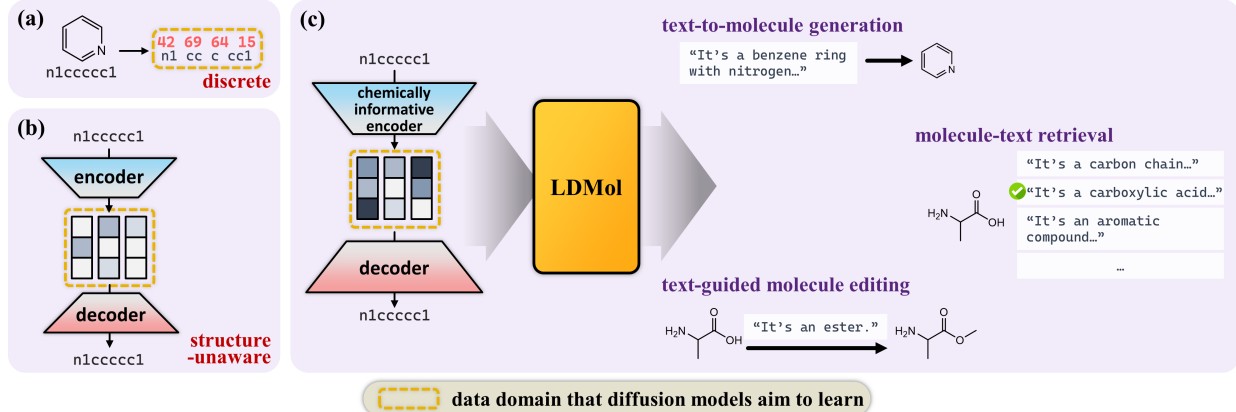

*Figure 1.* Different strategies of data domain selection for molecule diffusion models. (a) The model directly learns the raw representation of the molecule such as string tokens. (b) An autoencoder can be employed to let the generative model learn its latent distribution. (c) A regularized, chemically pre-trained encoder can provide a latent space readily learnable by external generative models.

generative models were based on autoregressive models.

To overcome this gap, we suggest that a latent domain (Vahdat et al., 2021; Rombach et al., 2022) is essential to train effective diffusion models for complex molecule generation tasks. Moreover, beyond the limitation of the previous works (Xu et al., 2023) that mainly focused on resolving the discreteness with naive reconstruction loss (Figure 1-(b)), we report that a latent encoder extracting rich and refined information about the molecule structure can further improve the generative model performance (Figure 1-(c)). Specifically, we design a novel Latent Diffusion Molecular generative model (LDMol) for text-conditioned molecule generation, trained on the latent space of the separately pre-trained molecule encoder. By preparing an encoder to provide a chemically useful and interpretable feature space, our model can more easily connect the molecule data with the highly complicated condition of natural text. In the process, we suggest a novel contrastive encoder training strategy by minimizing mutual information between positive SMILES pairs to encode a unique structural characteristic.

Extensive experimental results show that LDMol can outperform many state-of-the-art autoregressive models and generate valid SMILES that meet the input text condition. Considering SMILES as a variation of text data, we report one of the first diffusion models that successfully surpassed autoregressive models in textual data generation. This may suggest the possibility of improving existing diffusion models (Lovelace et al., 2024) for natural language through careful design of the latent space. Furthermore, LDMol can leverage the learned score function and be applied to several multi-modal downstream tasks such as molecule-to-text retrieval and text-guided molecule editing, without additional task-specific training. We summarize the contribution of this work as follows:

- We propose a latent diffusion model LDMol for text-conditioned molecule generation to generate valid molecules that are better aligned to the text condition. This approach demonstrates the potential of generative models for chemical entities in a latent space.

- We report the importance of preparing a chemically informative latent space for the molecule latent diffusion model, and suggest a novel contrastive learning method to train an encoder that captures the molecular structural characteristic.

- LDMol outperforms the text-to-molecule generation baselines, and its modeled conditional score function enables the advanced attributes of diffusion models including various applications like molecule-to-text retrieval and text-guided molecule editing.

## 2. Background

**Diffusion generative models.** Diffusion models first define a forward process that perturbs the original data, and generates the data from the known prior distribution by the learned reverse process of the pre-defined forward process. Ho et al. (2020) fixed their forward process by gradually adding Gaussian noise to the data, which can be formalized as follows:

$$q(x_t|x_{t-1}) = \mathcal{N}(x_t; \sqrt{1-\beta_t}x_{t-1}, \beta_t I) \quad (1)$$

where $\beta_t, t = 1, \ldots, T$ is a noise schedule. This definition of forward process allows us to sample $x_t$ directly from $q(x_t|x_0)$ as follows, where $\alpha_t = 1 - \beta_t$ and $\overline{\alpha}_t = \prod_{i=1}^t \alpha_i$:

$$x_t = \sqrt{\overline{\alpha}_t}x_0 + \sqrt{1-\overline{\alpha}_t}\epsilon, \text{ where } \epsilon \sim \mathcal{N}(0, I) \quad (2)$$

The model $\epsilon_\theta$ learns the reverse process $p(x_{t-1}|x_t)$ by approximating $q(x_{t-1}|x_t)$ with a Gaussian distribution

$p_\theta(x_{t-1}|x_t) = \mathcal{N}(x_{t-1}; \mu_\theta(x_t, t), \sigma_t^2 I)$ where

$$\mu_\theta(x_t, t) = \frac{1}{\sqrt{\alpha_t}} \left( x_t - \frac{1 - \alpha_t}{\sqrt{1 - \overline{\alpha}_t}} \epsilon_\theta(x_t, t) \right) \quad (3)$$

which can be trained by minimizing the difference between $\epsilon$ and $\epsilon_\theta(x_t, t)$:

$$\theta^* = \arg\min_\theta \mathbb{E}_{x_0, t, \epsilon} ||\epsilon - \epsilon_\theta(x_t, t)||_2^2 \quad (4)$$

Once $\theta$ is trained, novel data can be generated with the learned reverse process $p_\theta$; starting from the random noise $x_T \sim \mathcal{N}(0, I)$, the output can be gradually denoised according to the modeled distribution of $p_\theta(x_{t-1}|x_t)$.

Various real-world data generation tasks require to generate data $x_0$ with a given condition $c$. To build diffusion models that can generate data from the conditional data distribution $q(x_0|c)$, the model that predicts the injected noise should also be conditioned by $c$.

$$\theta^* = \arg\min_\theta \mathbb{E}_{x_0, c, t, \epsilon} ||\epsilon - \epsilon_\theta(x_t, t, c)||_2^2 \quad (5)$$

After the success of diffusion models in the image and video domain, various works tried to build diffusion models to generate text data. While many suggested training diffusion models on text tokens (Austin et al., 2021), word embedding (Li et al., 2022), or text autoencoder latent space (Lovelace et al., 2024), their performance has been suboptimal compared to autoregressive models (Brown et al., 2020). We assume that this can be resolved with a better latent space design that reflects the characteristics of the data domain, and suggest a latent diffusion model that outperforms autoregressive models for textual data.

**Conditional molecule generation.** As a promising tool for many important chemical and engineering tasks like *de novo* drug discovery and material design, conditional molecule generation has been extensively studied with various models including recurrent neural network (RNN)s (Segler et al., 2017), bidirectional RNN (Grisoni et al., 2020), graph neural networks (Lim et al., 2020), and variational autoencoders (Gómez-Bombarelli et al., 2018; Lim et al., 2018). With the advent of large and scalable pre-trained models with transformers (Vaswani et al., 2017), the controllable conditions became more abundant and complicated (Bagal et al., 2021; Chang & Ye, 2024). Recent works reached a text-guided molecule generation (Edwards et al., 2022; Su et al., 2022; Liu et al., 2024a) leveraging a deep comprehension ability for natural language, especially with recent emergence of large language model (LLM)s (Liu et al., 2024b).

Recent works attempted to import the success of the diffusion model into molecule generation. Several graph-based and point cloud-based works have built conditional diffusion models that could generate molecules with simple chemical and biological conditions (Hoogeboom et al., 2022; Luo et al., 2023; Trippe et al., 2023). Gong et al. (2024) attempted a text-conditioned molecule diffusion model trained on the sequence of tokenized SMILES indices. However, these models treated discrete molecules with continuous Gaussian diffusion, introducing arbitrary numeric values and suboptimal performances. Xu et al. (2023) employed an autoencoder to build a diffusion model on a smooth latent space, but its controllable conditions were still limited to several physiochemical properties.

## 3. Methods

In this section, we explain the overall model architecture and training procedure of the proposed LDMol, which are briefly illustrated in Figure 2.

### 3.1. Extracting structure-aware SMILES latent space

The primary goal of introducing autoencoders for image latent diffusion models is to map raw images into a low-dimensional space, which reduces the computation cost (Vahdat et al., 2021; Rombach et al., 2022). This is plausible because a high-resolution image has an enormous dimension in the pixel domain, yet each pixel contains little information.

In this work for molecule generation, we utilize a string-based notation SMILES, one of the most popular molecule representations in text-molecule pair databases and benchmarks. We built a SMILES encoder to map raw SMILES strings into a latent vector. In this case, the role of our SMILES autoencoder has to be different from that of the autoencoders for images; a molecule structure can be fully expressed by only a sequence of $L$ integers for SMILES tokens, where $L$ is the maximum token length. However, each token carries significant information, and hidden interactions between these tokens are much more complicated than interactions between image pixels. Therefore, the SMILES encoder should focus more on extracting chemical meaning into the latent space, even if it results in a latent space with more dimensions than the raw SMILES string.

A number of molecule encoders (Wang et al., 2019; Liu et al., 2024a; Zeng et al., 2022; Liu et al., 2023a) have been presented that can extract various useful chemical features, including biochemical activity or human-annotated descriptions. Nonetheless, these molecule encoders aim to extract certain desired features rather than encode all the information about the molecule structure. Therefore, the input cannot be fully restored from the model output.

Although autoencoders with appropriate regularization (*e.g.*, KL-divergence loss (Kingma, 2014; Gómez-Bombarelli et al., 2018)) provide a continuous and reconstructible

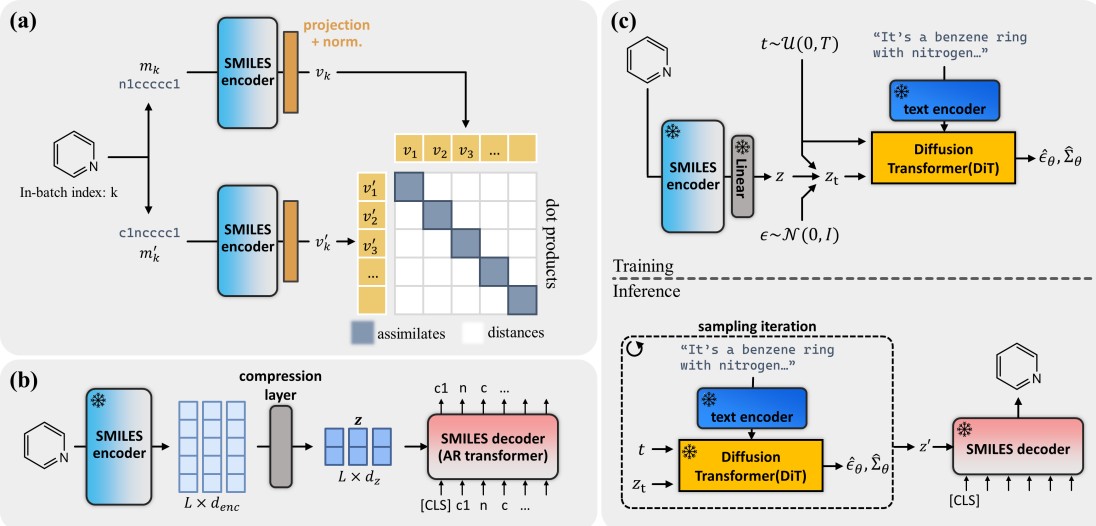

*Figure 2.* Overview of the proposed molecule autoencoder and the latent diffusion model. (a) SMILES encoder is trained with contrastive learning to construct latent space that embeds a structural characteristic. (b) After the SMILES encoder is prepared, a linear compression layer and an autoregressive decoder are trained to restore the encoder input. (c) The training and inference process of the latent diffusion model is conditioned by the output of the frozen external text encoder.

molecular latent space, their encoder output is not guaranteed to possess the characteristic of the underlying molecular structure, beyond the minimal information to reconstruct the input string. To visualize this, we prepared a trained $\beta$-VAE (Higgins et al., 2017) and measured the feature distance between two SMILES from the same molecule obtained via SMILES enumeration (Bjerrum, 2017). Here, SMILES enumeration is the process of writing out all possible SMILES of the same molecule, as illustrated in Figure 3-(a). Figure 3-(b) shows that $\beta$-VAE had difficulties assimilating features from the same molecule compared to the one between random SMILES pairs, indicating that it couldn't capture the intrinsic features beneath the SMILES string. This inconsistency makes it difficult for later models that learn this latent space to figure out the connection between the latent and the molecule, which could eventually degrade the performance as the condition gets more complex like natural texts. Assuming most of the controllable conditions has unavoidable correlation with the molecule structure, we insist that latent domain where the feature proximity is more structurally meaningful would benefit the conditional generative model.

Accordingly, here we propose three conditions that our SMILES autoencoder's latent space has to satisfy: enable reconstruction of the input, have as small dimensions as possible, and embed molecular structural information that can be readily learned by diffusion models.

**Encoder design.** In this respect, we train our SMILES encoder with contrastive learning (Figure 2-(a)), which aims to learn better representation by assimilating features containing similar information (*i.e.* positive pair) and distancing semantically unrelated features (*i.e.* negative pair). We define two enumerated SMILES from the same molecule as a positive pair and two SMILES from different molecules as a negative pair.

Here, we argue that the proposed contrastive learning with SMILES enumeration can train the encoder to encapsulate the unique structural characteristics of the input molecule: Contrastive learning learns an invariant for the augmentations applied on positive pairs (Zhang & Ma, 2022), and it is known that a good augmentation should reduce as much mutual information between positive pairs as possible while preserving relevant information (Tian et al., 2020). Meanwhile, enumerated SMILES of the same molecule are obtained by traversing the nodes and edges in the molecular graph with a different visiting order. Therefore, to detect all possible enumerated SMILES and find SMILES-enumeration-invariant, the model has to understand the entire connectivity between atoms. This makes the encoder output a unique characteristic that captures the overall molecular structure. Compared to the hand-crafted augmentations previously presented for molecule contrastive learning (You et al., 2020), enumerated SMILES pairs have minimal mutual information since we utilize all possible variations in the SMILES format. And since all enumerated SMILES are guaranteed to represent an identical molecule, there is no relevant information loss during the augmentation. Figure 3-(a) and Appendix B.1 demonstrate that our LDMol trained with contrastive learning with SMILES enumeration now correctly assimilates features from the same molecule, and its latent space cap-

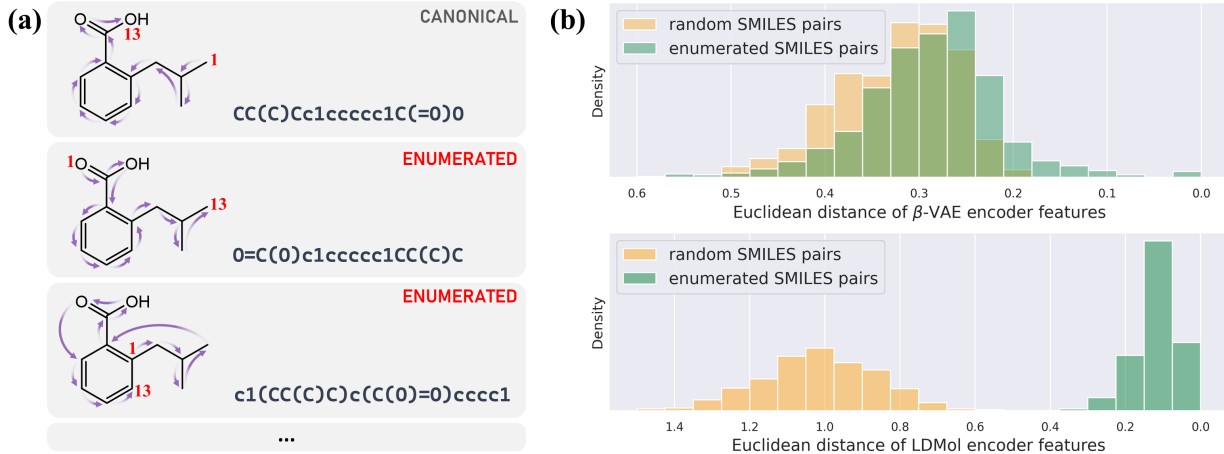

*Figure 3.* Behaviour of encoder features on SMILES enumeration. (a) Examples of SMILES enumeration with node traversal order. (b) Euclidean distance between features from $\beta$-VAE ($\beta = 0.001$) and our proposed encoder, with 1,000 random SMILES pairs and 1,000 enumerated SMILES pairs. The distance was rescaled by $1/\sqrt{d}$ where $d$ is a latent dimension size.

tures meaningful structural information of the molecule.

Specifically, a SMILES string is fed into the SMILES encoder $\mathcal{E}(\cdot)$ with a special "[SOS]" token which denotes the start of the sequence. For a batch of $N$ input SMILES $M = \{m_1, m_2, \ldots, m_k, \ldots, m_N\}$, we prepare a positive pair SMILES $m'_k$ for each $m_k$ by SMILES enumeration to construct $M' = \{m'_1, m'_2, \ldots, m'_k, \ldots, m'_N\}$. After $M$ and $M'$ are passed through the SMILES encoder, we feed each SMILES' output vector corresponding to the [SOS] token into an additional linear projection and normalization layer, denoting its output as $v_k$ and $v'_k (k = 1, 2, \ldots, N)$. Assimilating $v_k$ and $v'_k$ from the positive pairs and distancing the others can be done by minimizing the following InfoNCE loss (Oord et al., 2018).

$$\mathcal{L}_{con}(M, M') = -\sum_{k=1}^{N} \log \frac{\exp(v_k \cdot v'_k / \tau)}{\sum_{i=1}^{N} \exp(v_k \cdot v'_i / \tau)} \quad (6)$$

Here, $\tau$ is a positive temperature parameter. To utilize a symmetric loss against the input, we trained our encoder with the following loss function.

$$\mathcal{L}_{enc}(M, M') = \mathcal{L}_{con}(M, M') + \mathcal{L}_{con}(M', M) \quad (7)$$

**Compressing the latent space.** The pre-trained SMILES encoder maps a molecule into a vector of size $[L \times d_{enc}]$, where $d_{enc}$ is the feature size of the encoder. To avoid the curse of dimensionality and construct a more learnable feature space for diffusion models, we additionally employed a linear compression layer $f(\cdot)$ (Figure 2-(b)) to reduce the dimension from $[L \times d_{enc}]$ to $[L \times d_z]$. Here, we build our compression layer as simply as possible to prevent its output from deviating from the previous structure-aware and regulated features (See A.2 for further justification). The

range of this linear layer output is a target domain of our latent diffusion model.

**Decoder design.** When a SMILES $m$ is passed through the SMILES decoder and the compression layer, the SMILES decoder reconstructs $m$ from $f(\mathcal{E}(m))$. Note that the decoder knows nothing about SMILES distribution or its correlation with natural texts, and any design would be acceptable as long as it recovers SMILES from its latent. Following many major works that treated SMILES as a variant of language data (Segler et al., 2017; Chang & Ye, 2024), we built an autoregressive transformer (Vaswani et al., 2017) as our decoder (Figure 2-(b)) which is widely used to successfully generate sequential data with varied length (Brown et al., 2020). Specifically, starting from the [SOS] token, the decoder predicts the next SMILES token using information from $f(\mathcal{E}(m))$ with cross-attention layers. When $\{t_0, t_1, \ldots, t_n\}$ is the token sequence of $m$, the decoder is trained to minimize the next-token prediction loss described as Eq. (8). Here, the decoder and the compression layer are jointly trained while the encoder's parameter is frozen. After being fully trained, the decoder was able to reconstruct roughly 98% of the SMILES encoder input.

$$\mathcal{L}_{dec} = -\sum_{i=1}^{n} \log p(t_n | t_{0:n-1}, f(\mathcal{E}(m))) \quad (8)$$

### 3.2. Text-conditioned latent diffusion model

As shown in Figure 2-(c), our diffusion model learns the conditional distribution of the SMILES latent $z$ whose dimension is $[L \times d_z]$. In the training phase, a molecule in the training data is mapped to the latent $z$ and applied a forward noising process into $z_t$ with randomly sampled diffusion

timestep $t$ and injected noise $\epsilon$. A diffusion model predicts the injected noise from $z_t$, conditioned by the paired text description via a frozen external text encoder. In the inference phase, the diffusion model iteratively generates a new latent sample $z'$ from a given text condition, which is then decoded to a molecule via the SMILES decoder.

Since most contributions to diffusion models were made in the image domain, most off-the-shelf diffusion models have the architecture of convolution-based Unet (Ronneberger et al., 2015). However, introducing the spatial inductive bias of Unet cannot be justified for the latent space of our encoder. Therefore we employed DiT (Peebles & Xie, 2023) architecture, one of the most successful approaches to transformer-based diffusion models for more general data domain. Specifically, we utilized a DiT$_{base}$ model with minimal modifications to handle text conditions with cross-attention, where more details can be found in Section A.1.

Text-based molecule generation requires a text encoder to process natural language conditions. Existing text-based molecule generation models trained their text encoder from scratch (Pei et al., 2023), or utilize a separate encoder model pre-trained on scientific domain corpora (Beltagy et al., 2019). In this work, we took the encoder part of MolT5$_{large}$ (Edwards et al., 2022) as our text encoder.

### 3.3. Implementation details

The pre-training of the SMILES encoder and the corresponding decoder was done with 10,000,000 general molecules from PubChem (Kim et al., 2023). The SMILES tokenizer vocabulary consists of 300 tokens, which were obtained from the pre-training data SMILES corpus using the BPE algorithm (Gage, 1994). We only used SMILES data that does not exceed a fixed maximum token length $L$. To ensure enough batch size for negative samples (He et al., 2020), we build a memory queue that stores $Q$ recent input and use them for the encoder training. We found that if the training data have a stereoisomer, considering it as "hard-negative" samples and including it in the loss calculation batch helps the encoder training to differentiate different stereoisomers.

To train the text-conditioned latent diffusion model, we gathered three existing datasets of text-molecule pairs: PubchemSTM curated by Liu et al. (2023a), ChEBI-20 (Edwards et al., 2021), and PCdes (Zeng et al., 2022). Only a train split for each dataset was used for the training, and pairs that appeared in the test set for the experiments are additionally removed. We also used 10,000 molecules from ZINC15 (Sterling & Irwin, 2015) without any text descriptions, which helps the model learn the common distribution of molecules. When these unlabeled data were fed into the training model, we used a pre-defined null text for the absence of text condition. We utilized 320,000 training data in total, much smaller than recent transformer-based baselines (Edwards et al., 2022; Pei et al., 2024) with millions of unimodal and multimodal data from various databases.

The latent diffusion model was trained with the training loss suggested by Dhariwal & Nichol (2021). To take advantage of classifier-free guidance (Ho & Salimans, 2021), we randomly replaced 3% of the given text condition with the null text during the training. The sampling iteration in the inference stage used DDIM-based (Song et al., 2021a) 100 sampling steps with a classifier-free guidance. More detailed training hyperparameters can be found in Appendix A.2. The code for LDMol training and text-to-molecule sampling is available at https://github.com/jinhojsk515/LDMol.

## 4. Experiments

### 4.1. Text-conditioned molecule generation

In this section, we evaluated the trained LDMol's ability to generate molecules that agree with the given natural language conditions. First, we generated molecules with LDMol using the text captions in the ChEBI-20 test set and compared them with the ground truth. The metrics we've used are as follows: SMILES validity, BLEU score (Papineni et al., 2002) and Levenshtein distance between two SMILES, Tanimoto similarity (Bajusz et al., 2015) between two SMILES with three different fingerprints (MACCS, RDK, Morgan), the exact match ratio, and Frechet ChemNet Distance (FCD) (Preuer et al., 2018). We tested different scales for the classifier-free guidance scale $\omega$ in the sampling process and found $\omega = 2.5$ works best (See Section B.2).

Table 1 contains the performance of LDMol and other baselines for text-to-molecule generation on the ChEBI-20 and PCDes test set. Including both autoregressive models and diffusion-based models, LDMol outperformed the existing models in almost every metric. While few models showed higher validity than ours, they showed a lower agreement between the output and the ground truth, which we insist is a more important role of generative text-to-molecule models. Also, MolT5$_{large}$ uses the same text encoder as LDMol, yet there's a significant performance difference between the two models. We believe this is because our continuous and structure-aware latent space is much easier to learn and align with the same textual information, compared to the raw token sequence for transformer-based models.

To demonstrate the LDMol's molecule generalization ability with more broad and general text inputs, we analyzed the generated output with several hand-written prompts. These input prompts were not contained in the training data and were relatively vague and high-level so that many different molecules could satisfy the condition. Figure 4 shows the samples of generated molecules from LDMol with several input prompt examples. We found that LMDol can generate molecules with high validity that follow the various levels of

*Table 1.* Benchmark results of text-to-molecule generation on ChEBI-20 and PCDes test set. The best performance for each metric was written in **bold.** The "Family" column denotes whether the model is AR(autoregressive model) or DM(diffusion model).

| Dataset | Model | Family | Validity↑ | BLEU↑ | Levenshtein↓ | MACCS FTS↑ | RDK FTS↑ | Morgan FTS↑ | Match↑ | FCD↓ |
|---|---|---|---|---|---|---|---|---|---|---|
| | Transformer | AR | 0.906 | 0.499 | 57.660 | 0.480 | 0.320 | 0.217 | 0.000 | 11.32 |
| | GIT-Mol (Liu et al., 2024a) | AR | 0.928 | 0.756 | 26.315 | 0.738 | 0.582 | 0.519 | 0.051 | - |
| | T5$_{base}$ (Raffel et al., 2020) | AR | 0.660 | 0.765 | 24.950 | 0.731 | 0.605 | 0.545 | 0.069 | 2.48 |
| | MolT5$_{base}$ (Edwards et al., 2022) | AR | 0.772 | 0.769 | 24.458 | 0.721 | 0.588 | 0.529 | 0.081 | 2.18 |
| | T5$_{large}$ | AR | 0.902 | 0.854 | 16.721 | 0.823 | 0.731 | 0.670 | 0.279 | 1.22 |
| ChEBI-20 | MolT5$_{large}$ | AR | 0.905 | 0.854 | 16.071 | 0.834 | 0.746 | 0.684 | 0.311 | 1.20 |
| | MolXPT (Liu et al., 2023b) | AR | 0.983 | - | - | 0.859 | 0.757 | 0.667 | 0.215 | 0.45 |
| | bioT5 (Pei et al., 2023) | AR | **1.000** | 0.867 | 15.097 | 0.886 | 0.801 | 0.734 | 0.413 | 0.43 |
| | bioT5+ (Pei et al., 2024) | AR | **1.000** | 0.872 | 12.776 | 0.907 | 0.835 | 0.779 | 0.522 | 0.35 |
| | TGM-DLM (Gong et al., 2024) | DM | 0.871 | 0.826 | 17.003 | 0.854 | 0.739 | 0.688 | 0.242 | 0.77 |
| | LDMol | DM | 0.941 | **0.926** | **6.750** | **0.973** | **0.950** | **0.931** | **0.530** | **0.20** |
| | MolT5$_{large}$ | AR | 0.944 | 0.692 | 18.481 | 0.810 | 0.741 | 0.699 | 0.440 | 0.70 |
| PCDes | bioT5 | AR | **1.000** | 0.754 | 15.658 | 0.797 | 0.726 | 0.677 | 0.455 | 0.69 |
| | bioT5+ | AR | 0.999 | 0.677 | 20.464 | 0.743 | 0.615 | 0.541 | 0.266 | 1.09 |
| | LDMol | DM | 0.944 | **0.857** | **8.726** | **0.885** | **0.817** | **0.780** | **0.464** | **0.32** |

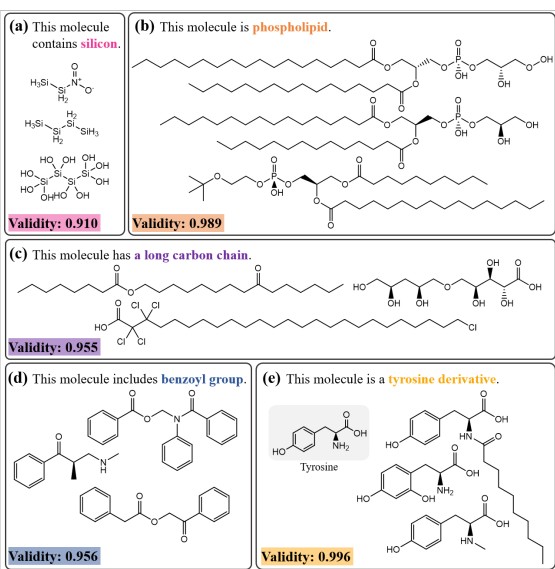

(a) This molecule contains silicon. Validity: 0.910
(b) This molecule is phospholipid. Validity: 0.989
(c) This molecule has a long carbon chain. Validity: 0.955
(d) This molecule includes benzoyl group. Validity: 0.956
(e) This molecule is a tyrosine derivative. Validity: 0.996

*Figure 4.* Examples of the generated molecules by LDMol with various text conditions, with validity on 1,000 generated samples.

input conditions for specific atoms(a), compound class(b), molecular substructure(c), functional groups(d), and substance names(e). The validity was calculated by the number of valid SMILES over 1,000 generated samples, above 0.9 for most scenarios we tested. Considering that these short, broad, and hand-written text conditions are distinct from the text conditions in the training dataset, we've concluded that our model is able to learn the general relation between natural language and molecules. We conducted quantitative analyses on the case studies and additional examples with hand-written prompts, which can be found in Appendix B.3.

### 4.2. Applications toward downstream tasks

Well-trained diffusion models learned the score function of a data distribution, which enables high applicability to various

*Table 2.* 64-way accuracy in % on molecule-to-text retrieval task. For LDMol, $n$ is a number of iterations where $||\hat{\epsilon}_\theta - \epsilon||_2^2$ was calculated. The best performance for each task is written in **bold.**

| Model | PCdes test set | | MoMu test set | |
|---|---|---|---|---|
| | sentence | paragraph | sentence | paragraph |
| SciBERT (Beltagy et al., 2019) | 50.4 | 82.6 | 1.38 | 1.38 |
| KV-PLM (Zeng et al., 2022) | 55.9 | 77.9 | 1.37 | 1.51 |
| MoMu-S (Su et al., 2022) | 58.6 | 80.6 | 39.5 | 45.7 |
| MoMu-K | 58.7 | 81.1 | 39.1 | 46.2 |
| MoleculeSTM (Liu et al., 2023a) | - | 81.4 | - | 67.6 |
| MolCA (Liu et al., 2024c) | - | 86.4 | - | 73.4 |
| LDMol($n$=10) | 60.7 | 90.2 | 66.4 | 84.8 |
| LDMol($n$=25) | **62.2** | **90.3** | **78.4** | **87.1** |

downstream tasks. The state-of-the-art image diffusion models have shown their versatility in image editing (Meng et al., 2022; Hertz et al., 2023), classification (Li et al., 2023), retrieval (Jin et al., 2023), inverse problems like inpainting and deblurring (Chung et al., 2023), image personalization (Ruiz et al., 2023), etc. To demonstrate LMDol's potential versatility as a diffusion model, we applied the pre-trained LDMol to the molecule-to-text retrieval and text-guided molecule editing. See Appendix A.3 for a more detailed procedure for each downstream task.

**Molecule-to-text retrieval.** Our approach to molecule-to-text retrieval is similar to the idea of using a pre-trained diffusion model as a classifier (Li et al., 2023): LDMol takes each candidate text with a query molecule's noised latent, and retrieves the text that minimizes the noise estimation error $||\hat{\epsilon}_\theta - \epsilon||_2^2$ between the injected noise $\epsilon$ and the predicted noise $\hat{\epsilon}_\theta$. Since this process has randomness due to the stochasticity of $t$ and $\epsilon$, we repeated the same process $n$ times with resampled $t$ and $\epsilon$ and used a mean error to minimize the performance variance.

We measured a 64-way in-batch retrieval accuracy of LD-Mol using two different test sets: PCdes test split and MoMu retrieval dataset curated by Su et al. (2022), where the result with other baseline models are listed in Table 2. Only one

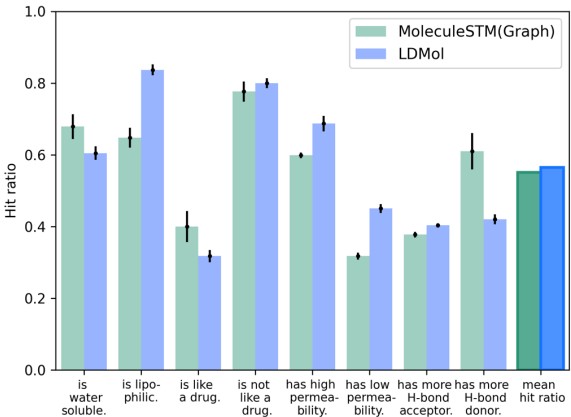

*Figure 5.* Hit ratio of molecule editing by LDMol and MoleculeSTM (Liu et al., 2023a) in eight scenarios. In the figure, we omitted "*This molecule*" in front of the actual prompts, and abbreviated "*hydrogen bonding*" to "*H-bond*".

randomly selected sentence in each candidate description was used for the retrieval in the "sentence" column, and all descriptions were used for the "paragraph" column. LDMol achieved a higher performance in all four scenarios compared to the previously presented models and maintained its performance on a relatively out-of-distribution MoMu test set with minimal accuracy drop. LDMol became more accurate as the number of function evaluations increased, and the improvement was more significant at the sentence-level retrieval and out-of-distribution dataset. The actual examples from the retrieval result can be found in Appendix B.4.

**Text-guided molecule editing.** We applied a method of Delta Denoising Score (DDS) (Hertz et al., 2023), which was originally suggested for text-guided image editing, to see whether LDMol can be used to optimize a source molecule to match a given text. Using two text prompts that describe the source data $z_{src}$ and the desired target, DDS presents how a text-conditioned diffusion model can modify $z_{src}$ into a new data $z_{tgt}$ that follows the target text prompt.

We imported a method of DDS on LDMol's molecule latent to edit a given molecule to match the target text, with several prepared editing prompts that require the model to change certain atoms, substructures, and intrinsic properties from the source molecule. Figure 5 shows that LDMol had comparable performance with a previously suggested text-guided molecule editing model (Liu et al., 2023a), with a higher hit ratio in five out of eight scenarios. Several editing examples with hand-written scenarios are shown in Appendix B.5.

### 4.3. Effectiveness of the suggested latent space

We've conducted an ablation study in Table 3 that compares LDMol with latent diffusion models trained on naively con-

*Table 3.* Quantitative results of the ablation study. The best performance for each metric is written in **bold.**

| models | Autoencoder | ChEBI20 generation | | |
|---|---|---|---|---|
| | Recon. Acc.↑ | Validity↑ | Match↑ | FCD↓ |
| LDMol w/o contrastive learning | **1.000** | 0.019 | 0.000 | 58.60 |
| LDMol w/ $\beta$-VAE ($\beta$=0.001) | 0.999 | 0.847 | 0.492 | 0.34 |
| LDMol | 0.983 | **0.941** | **0.530** | **0.20** |

structed latent space, to emphasize the benefit of the suggested encoder training. Each model is pre-trained with the same number of DiT training iterations. We've also performed an ablation study on more detailed design choices of LDMol, which can be found in Appendix B.6.

When we remove the contrastive encoder pre-training objective and construct the molecule latent space with a naive autoencoder, the diffusion model completely fails to learn the latent distribution to generate valid SMILES. On the other hand, a $\beta$-VAE with KL-divergence regularization has reconstructible latent space and showed a text-to-molecule generation match ratio of 0.492, which already outperforms the previous diffusion model TGM-DLM and several autoregressive models in Table 1. This demonstrates the necessity of diffusion models in the continuous data domain, with their potential to be successful on discrete molecule data comparable to the autoregressive models. Nonetheless, its overall metric is still worse than the proposed LDMol, with notably low validity and FCD. We insist that this gap comes from the structurally informative latent space of LDMol which is easier for the diffusion model to learn the correlation between the latent space and the condition.

## 5. Conclusion

In this work, we presented a text-to-molecule diffusion model LDMol that runs on a chemical latent space reflecting structural information. By introducing the deeply studied paradigm of the latent diffusion model with carefully designed latent encoder, LDMol retains many advanced attributes of diffusion models that enable various applications.

Despite the noticeable performances of LDMol, it still has limitations that can be improved, as LDMol still often struggles to follow some text conditions such as complex biological properties. Nonetheless, we expect that the LDMol's performance could be improved further with the emergence of richer text-molecule pair data and more powerful text encoders. Moreover, combining physiochemical and biological annotations on top of the structurally informative latent space is a promising future work that can ease the connection between molecules and text conditions.

We believe that our approach could inspire tackling various chemical generation tasks using latent space, not only text-conditioned but also many more desired properties, such as

biochemical activity. Especially, we expect LDMol to be a starting point to bridge achievements in the state-of-the-art diffusion model into the chemical domain.

## Acknowledgments

This work was supported by the National Research Foundation of Korea under Grant No. RS-2024-00336454, and the Institute of Information & Communications Technology Planning & Evaluation(IITP) grant funded by the Korea government(MSIT) (RS-2025-02304967, AI Star Fellowship(KAIST)).

## Impact Statement

This paper presents work whose goal is to advance the field of machine learning and its chemical applications. There are many potential societal consequences of our work, including the possibility of the molecular design for harmful or inappropriate properties.

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

# A. Experimental details

## A.1. DiT block for SMILES latent diffusion model with text conditions

The DiT block architecture of the class-conditioned image diffusion model published by Peebles & Xie (2023) is shown in Figure 6-(a). The noised input image latent $z_t$ is passed through a patch embedding layer and spatially flattened to be fed into the DiT block. The condition embedding $y$ and diffusion timestep embedding $t$ are incorporated into the model prediction via adaptive layer norm. The dimension of $t$ and $y$ are both $[B \times F]$, where $B$ is the batch size and $F$ is the number of features.

In the case of LDMol, the input latent $z_t$ with dimension $[B \times L \times F]$ is already spatially one-dimensional, we simply pass it through a linear layer to prepare DiT block input. Also, the text condition feature we've used has a much higher dimension of $[B \times L' \times F]$ where $L'$ is the token length of the text condition. Therefore, we stacked a cross-attention layer for text condition features after each self-attention layer, as shown in Figure 6-(b).

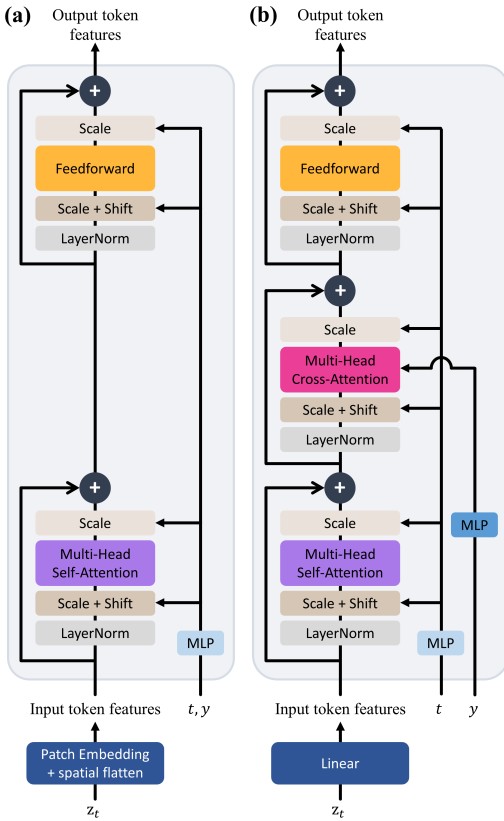

*Figure 6.* Input embedding layer and DiT blocks in the (a) originally published DiT and (b) LDMol.

## A.2. Model hyperparameters and training setup

The LDMol encoder and decoder consist of 12 transformer layers of $BERT_{base}$, where the decoder has a causal mask in its self-attention layers and includes a cross-attention layer after each self-attention layer to receive latent information. Detailed hyperparameters on the model architecture are listed in Table 4, with settings on the model training procedure. Here, we provide the results of several ablation studies to support our selection of the hyperparameters.

It is known that lower temperature $\tau$ in contrastive learning penalizes hard negatives more strongly, making the learned feature more sensitive to fine-grain details (Wang & Liu, 2021). We considered this as a desirable property for our latent space and used a small tau of 0.07. When we used too big $\tau$ of 0.15 as shown in Figure 7, it reduced the model's ability to distinguish different molecules and made the training loss converge to a much higher value.

Table 5 lists the LDMol autoencoder's SMILES reconstruction accuracy in various autoencoder training strategies. When we

*Table 4.* The choice of the model hyperparameters and training setup.

| hyperparameters | |
|---|---|
| $L$ | 128 |
| $d_{enc}$ | 1024 |
| $d_z$ | 64 |
| $\tau$ | 0.07 |
| $Q$ | 16384 |
| **training setup** | |
| optimizer | autoencoder: AdamW, DiT: Adam |
| learning rate | autoencoder: cosine annealing(1e-4→1e-5), DiT: 1e-4 |
| batch size per GPU | encoder: 64, decoder: 128, DiT: 64 |
| training resources | 8 NVIDIA A100(VRAM:40GB) |

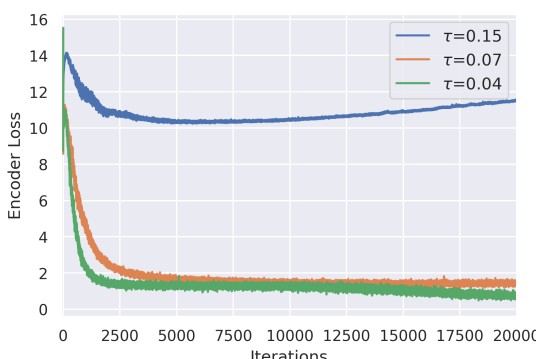

*Figure 7.* The encoder loss convergence with different temperature parameter $\tau$.

| Methods | | Recon. Acc. |
|---|---|---|
| Contrastive learning on compressed latent with $d_z$=64 | | 0.084 |
| Contrastive learning on $d_{enc}$=1024 | compression with $d_z$=32 | 0.948 |
| | compression with $d_z$=64 | 0.980 |
| | compression with $d_z$=128 | 0.989 |

*Table 5.* SMILES reconstruction accuracy of various trained autoencoders, with 1,000 unseen SMILES.

apply contrastive loss directly into the compressed latent domain, the encoder fails to capture informative features, makes the decoder couldn't reconstruct the input molecule. In the scenario of adding linear compression after contrastive training with $d_{enc}$=1024, we observed an error rate of more than 5% for the compression size of $d_z$=32. Compression with $d_z$=128 slightly increased the reconstruction accuracy compared to $d_z = 64$, but the training time for the subsequent diffusion model rapidly increased. Considering that the failed 2% for the current model were mostly very long molecules, we concluded that $d_z = 64$ is sufficient for our model.

### A.3. LDMol's application on downstream tasks

---

**Algorithm 1** Molecule-to-Text Retrieval with LDMol

**Require:** $z, \mathcal{C} = \{c_i\}_{i=1}^{B}, n \in \mathbb{N}^+$
1: Initialize $\texttt{Errors}[c_i] = 0$ **for** $i = 1$ **to** $B$
2: **for** $\texttt{iter} = 1$ **to** $n$ **do**
3:    $t \sim U[0, T], \epsilon \sim \mathcal{N}(0, I)$
4:    $z_t = \sqrt{\overline{\alpha}_t} z + \sqrt{1 - \overline{\alpha}_t}\epsilon$
5:    **for** $i = 1$ **to** $B$ **do**
6:      $\texttt{Errors}[c_i] += ||\hat{\epsilon}_\theta(z_t, t, c_i) - \epsilon||_2^2$
7:    **end for**
8: **end for**
9: **Return** $\arg\min_{c_i \in \mathcal{C}} \texttt{Errors}[c_i]$

---

**Algorithm 2** Text-guided Molecule Editing with LDMol

**Require:** $z_{src}, c_{src}, c_{tgt}, N \in \mathbb{N}^+, \gamma > 0, \omega \geq 1, \mathcal{D}$
1: Initialize $z_{tgt} = z_{src}$
2: **for** $\texttt{iter} = 1$ **to** $N$ **do**
3:    $t \sim U[0, T], \epsilon \sim \mathcal{N}(0, I)$
4:    $z_{t,src} = \sqrt{\overline{\alpha}_t} z_{src} + \sqrt{1 - \overline{\alpha}_t}\epsilon$
5:    $z_{t,tgt} = \sqrt{\overline{\alpha}_t} z_{tgt} + \sqrt{1 - \overline{\alpha}_t}\epsilon$
6:    $\epsilon_{\theta,src}^\omega = (1 - \omega)\epsilon_\theta(z_{t,src}, t, \varnothing) + \omega\epsilon_\theta(z_{t,src}, t, c_{src})$
7:    $\epsilon_{\theta,tgt}^\omega = (1 - \omega)\epsilon_\theta(z_{t,tgt}, t, \varnothing) + \omega\epsilon_\theta(z_{t,tgt}, t, c_{tgt})$
8:    $z_{tgt} = z_{tgt} - \gamma(\epsilon_{\theta,tgt}^\omega - \epsilon_{\theta,src}^\omega)$
9: **end for**
10: **Return** $\mathcal{D}(z_{tgt})$

---

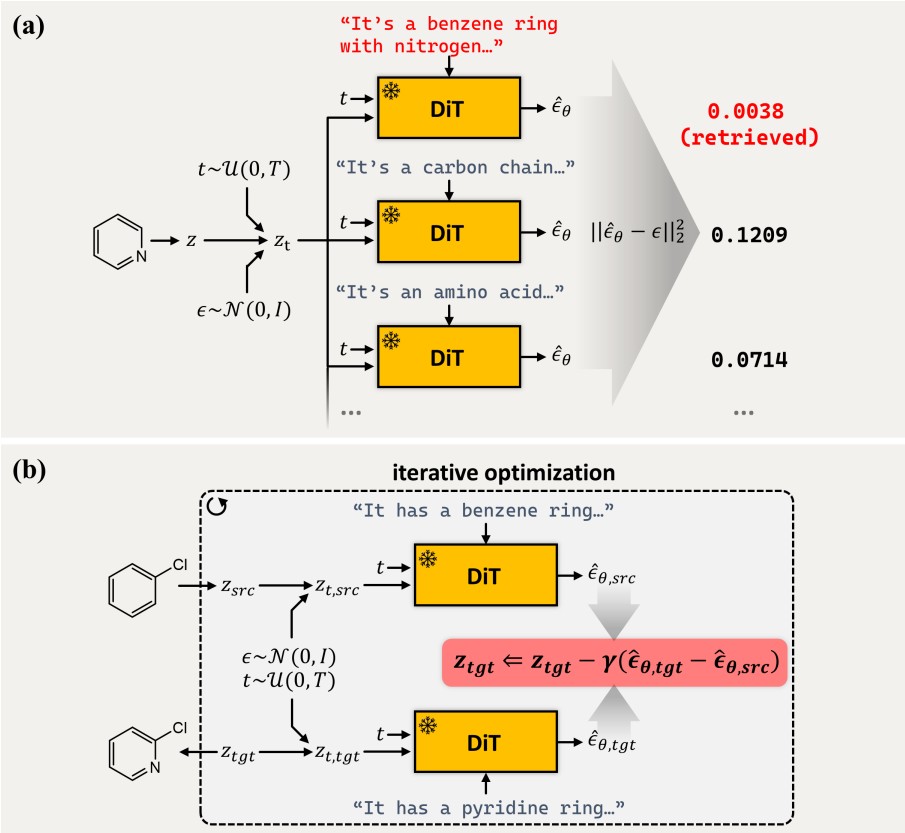

*Figure 8.* Overall pipeline for the downstream task applications of LDMol. (a) Molecule-to-text retrieval. (b) Text-guided molecule editing. The SMILES autoencoder and the text encoder are not drawn in this figure.

Figure 8-(a) and Algorithm 1 show the LDMol's molecule-to-text retrieval process with a given query molecule and text candidates $\mathcal{C} = \{c_i\}_{i=1}^B$. A given query molecule is converted to a latent $z$, and then a forward noise process is applied with a randomly sampled timestep $t$ and noise $\epsilon$. This $z_t$ is fed to LDMol with each candidate $c_i$, and the candidate that minimizes the loss $||\hat{\epsilon}_\theta(z_t, t, c_i) - \epsilon||_2^2$ between $\epsilon$ and the output noise $\hat{\epsilon}_\theta(z_t, t, c_i)$ is retrieved. To minimize the variance from the stochasticity of $t$ and $\epsilon$, the same process can be repeated $n$ times with resampled $t$ and $\epsilon$ to use a mean loss.

Figure 8-(b) and Algorithm 2 illustrate the DDS-based molecule editing with LDMol. Specifically, DDS requires source data $z_{src}$, target data $z_{tgt}$ which is initialized to $z_{src}$, and their corresponding source and target text descriptions $\{c_{src}, c_{tgt}\}$. We apply the forward noise process to $z_{src}$ and $z_{tgt}$ using the same randomly sampled $t$ and $\epsilon$ to get $z_{t,src}$ and $z_{t,tgt}$. These are fed into the pre-trained LDMol with their corresponding text, where we denote the output noise as $\hat{\epsilon}_{\theta,src}$ and $\hat{\epsilon}_{\theta,tgt}$. Finally, $z_{tgt}$ is modified towards the target text by optimizing it to the direction of $(\hat{\epsilon}_{\theta,tgt} - \hat{\epsilon}_{\theta,src})$ with a learning rate $\gamma$. Here, $\hat{\epsilon}_{\theta,tgt}$ and $\hat{\epsilon}_{\theta,tgt}$ can be replaced with the classifier-free-guided noises, utilizing the output with the null text and the guidance scale $\omega$. $z_{tgt}$ is decoded back as the editing output after the optimization step is iterated $N$ times. In Figure 5, where we applied the same scenario to a batch of molecules, we used null text as $c_{src}$ since it's impractical to prepare a source prompt for each molecule. The hyperparameters $\{N, \gamma, \omega\}$ are fixed for each scenario, where every choice is in the range of $100 \leq N \leq 200$, $0.1 \leq \gamma \leq 0.3$ and $2.0 \leq \omega \leq 4.5$. Following MoleculeSTM, each scenario was applied to 200 randomly sampled molecules from ZINC15, and the mean and standard deviation on three separate runs were plotted.

## B. Additional results

### B.1. Visualization of the LDMol latent space

To visualize the structural information encoded in the latent space of our encoder, we prepared 10 molecular clusters that contain 100 molecules, each sharing the common Murcko scaffold (Bemis & Murcko, 1996). Then, we obtained their

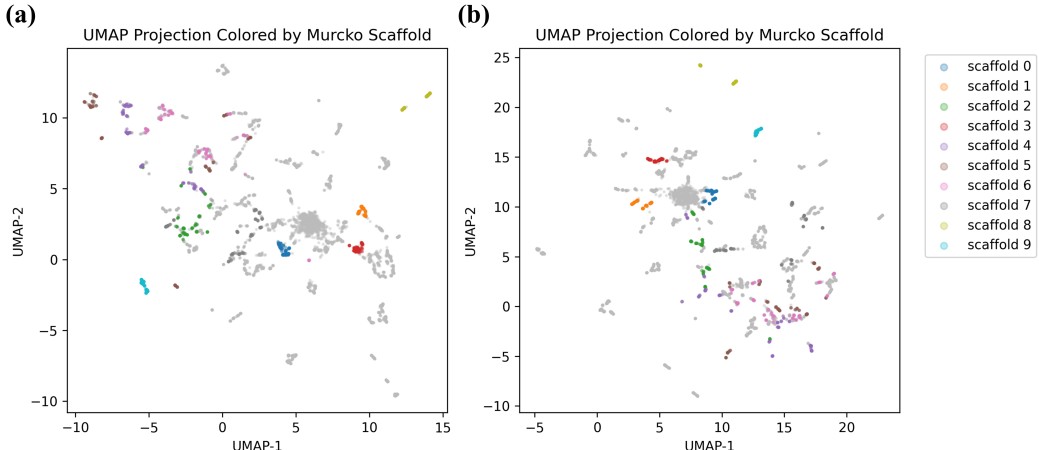

*Figure 9.* UMAP visualization of LDMol encoder output (a) and the final latent space after the linear compression layer (b), from 10 groups containing 100 molecules each with shared Murcko scaffold(colored) and 5,000 general molecules(light grey).

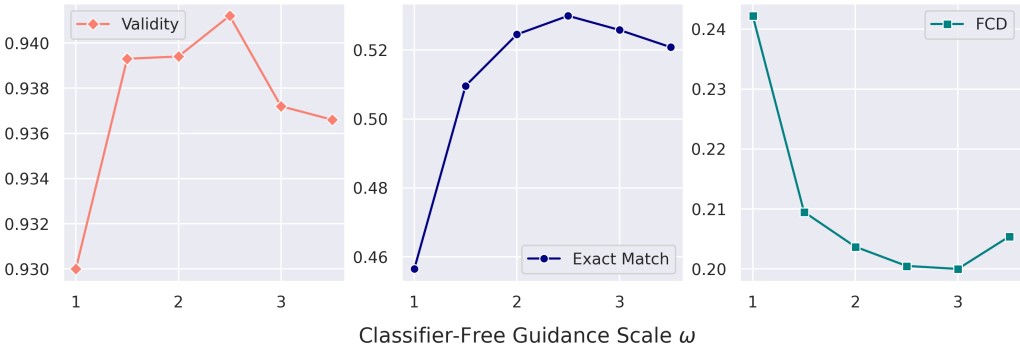

*Figure 10.* Text-to-molecule generation performance of LDMol against different classifier-free guidance scales.

latent vector from the LDMol encoder and visualized them in 2D via UMAP (McInnes et al., 2018). Note that As shown in Figure 9, the molecules with shared Murcko scaffold have formed clusters in the latent vector space.

## B.2. Effect of classifier-free guidance scale

Figure 10 plots the LDMol's text-to-molecule generation performance on the ChEBI-20 test set, with different classifier-free guidance scale $\omega$ in the sampling process. Starting from $\omega = 1.0$, which is equivalent to a naive conditional generation, we observed that the overall sample quality is improved as $\omega$ increases but collapses for too big $\omega$. This agrees with the well-known observation on image diffusion models, and we decided to use $\omega = 2.5$ for the text-to-molecule generation with LDMol.

## B.3. Text-to-molecule generation

We measured the uniqueness, novelty, and prompt alignment score for the prompts in Figure 4 using 1,000 samples. Validity is the proportion of generated SMILES that are valid. Uniqueness is the proportion of valid SMILES that are unique. The "align" score is the proportion of unique SMILES that match the given prompt. Novelty is the proportion of the unique SMILES that are not included in the training dataset. The alignment score was measured with SMILES pattern matching with the substructure described by the prompt. We observed that even when stochastic sampling was enabled, AR models struggled to generate various samples from a single prompt. LDMol can generate molecules that align better with various hand-written prompts. Furthermore, its outputs were much more diverse than the previous AR models.

Figure 11 shows the behavior of LDMol's text-to-molecule generation with several exceptional scenarios. When we fed

*Table 6.* Quantitative results of the case studies in Figure 4. The best performance for each metric is written in **bold.**

|  | Models | Validity(V) | Uniqueness(U) | Align(A) | V×U×A | Novelty |
|---|---|---|---|---|---|---|
| | molT5$_{large}$ | **0.996** | 0.006 | **1.000** | 0.006 | - |
| Case (a) | bioT5+ | 0.846 | 0.028 | 0.625 | 0.015 | - |
| | LDMol | 0.910 | **0.951** | **1.000** | **0.865** | 0.988 |
| | molT5$_{large}$ | 0.927 | 0.012 | 0.818 | 0.009 | - |
| Case (b) | bioT5+ | **1.000** | 0.573 | 0.782 | 0.448 | - |
| | LDMol | 0.989 | **0.960** | **0.906** | **0.860** | 0.958 |
| | molT5$_{large}$ | 0.783 | 0.072 | 0.643 | 0.036 | - |
| Case (c) | bioT5+ | **1.000** | 0.160 | **0.750** | 0.120 | - |
| | LDMol | 0.955 | **0.861** | 0.688 | **0.566** | 0.780 |
| | molT5$_{large}$ | 0.995 | 0.002 | 0.500 | 0.001 | - |
| Case (d) | bioT5+ | **1.000** | 0.015 | **0.733** | 0.011 | - |
| | LDMol | 0.956 | **0.849** | 0.703 | **0.571** | 0.842 |
| | molT5$_{large}$ | 0.956 | 0.015 | 0.571 | 0.008 | - |
| Case (e) | bioT5+ | **1.000** | 0.035 | 0.086 | 0.003 | - |
| | LDMol | 0.996 | **0.187** | **0.595** | **0.111** | 0.667 |

*Table 7.* Quantitative results of the ablation study. The best performance for each metric is written in **bold.**

| models | Autoencoder | ChEBI-20 text-to-molecule generation | | |
|---|---|---|---|---|
| | Recon. Acc.↑ | Validity↑ | Match↑ | FCD↓ |
| LDMol w/o compression layer | 0.964 | 0.022 | 0.000 | 67.93 |
| LDMol w/ transformer compression layer | **0.986** | 0.565 | 0.084 | 2.19 |
| LDMol w/o stereoisomer hard-negative | 0.891 | 0.939 | 0.278 | 0.24 |
| LDMol | 0.983 | **0.941** | **0.530** | **0.20** |

a completely ambiguous input such as *"beautiful"* or *"important"*, the model spits out a variety of different molecules without any consistency. When we fed contradictory inputs that could not be satisfied, the outputs were chimeric between contradictory prompts, with a clearly decreased validity.

### B.4. Molecule-to-text retrieval

Figure 13 contains examples of molecule-to-text retrieval results with molecules from the PCdes test set. The retrieval was done at the sentence level, and the top three retrieval outputs for each query molecule are described. The corresponding description from the data pair was correctly retrieved at first for all cases, and the other retrieved candidates show a weak correlation with the query molecule.

### B.5. Text-guided molecule editing

Figure 12 illustrates several case studies with hand-written editing prompts and results, where the editing output successfully modified the input molecule towards the target prompt with minimal corruption of the unrelated region. Here, we repeated DDS iterations with $N = 150$, $\gamma = 0.1$ and $\omega = 2.5$.

### B.6. Ablation study

We've conducted an ablation study on more detailed design choices of the proposed LDMol in Table 7 to analyze and emphasize their role.

When we didn't introduce a compression layer, the later diffusion model completely failed to learn the latent space since its dimension was too big for the diffusion model to learn. We tried to utilize a more complex compression module by transformer encoder layers of Perceiver-Resampler (Alayrac et al., 2022; Lovelace et al., 2024) manner, but the performance was significantly decreased as shown in the second row. This is presumably because adding another complicated layer makes the latent space deviate from the former informative and well-regulated learnable space.

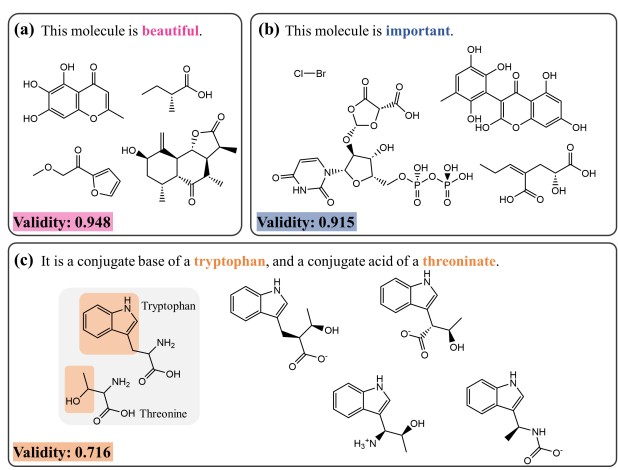

**(a)** This molecule is beautiful.

Validity: 0.948

**(b)** This molecule is important.

Validity: 0.915

**(c)** It is a conjugate base of a tryptophan, and a conjugate acid of a threoninate.

Validity: 0.716

This molecule has a pyrazine ring. →
This molecule has a pyrazine ring and acetyl group.

This molecule contains fluorine. →
This molecule contains bromine.

This molecule is a carboxylic acid. →
This molecule is an ester.

*Figure 11.* Examples of the generated molecules by LDMol, with (a, b) ambiguous text conditions and (c) contradictory and unreasonable input.

*Figure 12.* Examples of text-guided molecule editing with LDMol. The difference between the source text and the target text, and the corresponding region, is colored in purple.

When stereoisomers were not utilized as hard negative samples in the contrastive encoder training, the constructed latent space was not detailed enough to specify the input, which degraded the reconstruction accuracy of the autoencoder. The similarity metric of FCD didn't decrease as much, but the exact match ratio has decreased significantly.

### B.7. Computational efficiency

*Table 8.* Quantitative results of the ablation study.

| Models | molT5$_{large}$ | bioT5+ | LDMol |
|---|---|---|---|
| Required time[s] | 523 | 180 | 361 |
| VRAM usage[GB] | 4.92 | 1.08 | 3.79 |

Table 8 compares the computational efficiency of LDMol and several baselines with state-of-the-art performance. In terms of memory usage, our model can operate with less than 4GB of VRAM, which is smaller than that of molT5$_{large}$. The required time was also comparable to transformer-based models, even with the latent decoder and the Classifier-Free Guidance(CFG) which doubles the diffusion model usage. Considering many works have been published to reduce the inference time of diffusion models (Song et al., 2023; Salimans & Ho, 2022), as one of the first successful text-to-molecule diffusion models, we believe that the inference time can be further improved by future research.

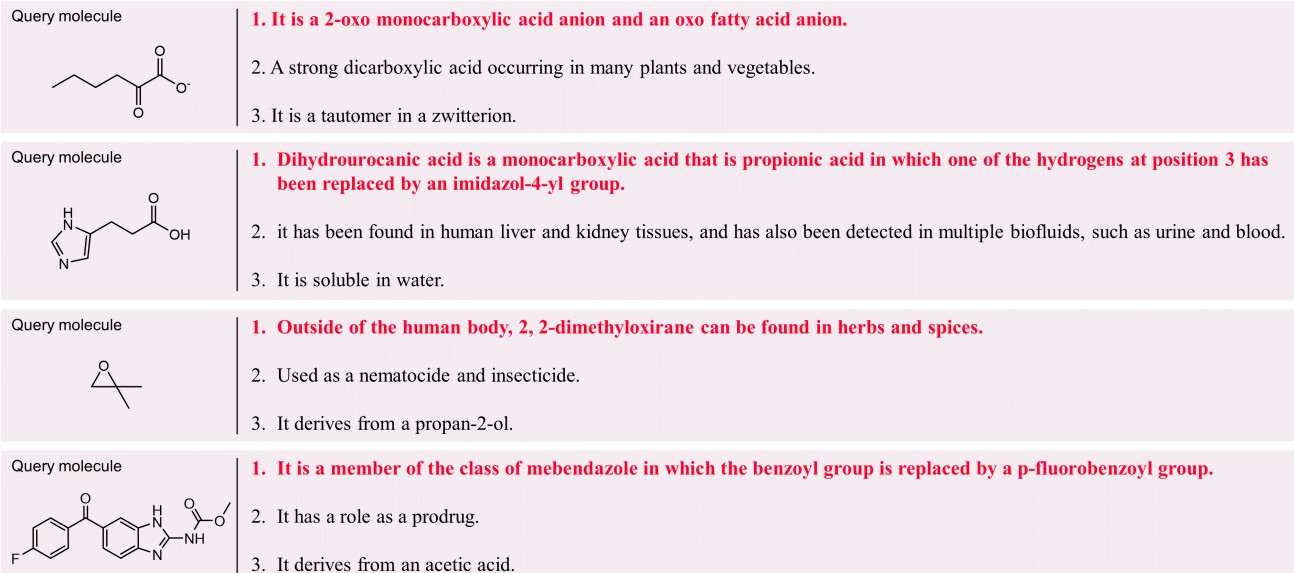

*Figure 13.* The examples of molecule-to-text retrieval result on the PCdes test set. Three sentences with the lowest noise estimation error were retrieved for each query molecule.

