# OpenReview forum: "LDMol: A Text-to-Molecule Diffusion Model with Structurally Informative Latent Space Surpasses AR Models"
_ICML.cc/2025/Conference — ICML 2025 poster_

### Official Review · Reviewer_mgY4 · 2025-03-03

**Overall Recommendation:** 3

**Summary:**

The authors introduce LDMol, a latent diffusion model for generating molecules based on text inputs. By integrating a chemically informed autoencoder and utilizing contrastive learning, their proposed diffusion model ensures structural consistency in the latent space, addressing the problem of multiple SMILES representations for the same molecule. Experimental results show that LDMol outperforms existing models in terms of SMILES validity, structural accuracy, and adherence to conditions. In addition, their LDMol is versatile, supporting tasks such as molecule-to-text retrieval and text-guided molecule editing, and provides a structurally aware diffusion approach as a compelling alternative to autoregressive models in molecular generation.

**Claims And Evidence:**

Yes

**Essential References Not Discussed:**

Yes

**Experimental Designs Or Analyses:**

Yes

**Methods And Evaluation Criteria:**

Yes

**Other Comments Or Suggestions:**

See above

**Other Strengths And Weaknesses:**

Strengths:
1) The proposed model shows its effectiveness in text-guided molecule generation, which highlighting its potential applications across different areas of chemical and biomedical research.
2) The source code is available, and the paper is well-written.
3) The experiments are evaluated using a variety of measures and experimental results are compared with several baseline models.

Weaknesses:
1) In addition to the existing metrics, it would be valuable to include other statistical measures, such as uniqueness and novelty, to provide a more comprehensive evaluation of the model's performance and its ability to generate innovative and diverse results.
2) The model demonstrates strong performance on the datasets, its ability to generalize to other types of chemical data or a wider range of text inputs warrants further investigation. Expanding the model's testing to diverse datasets and inputs would provide a clearer understanding of its versatility and robustness across different chemical and textual domains.
3) The generation of molecules from text descriptions has been explored in numerous other studies. It would be valuable to investigate whether the model can be extended to handle other types of data, such as molecular graphs or motifs. Expanding the model's capabilities to incorporate these additional data formats could significantly enhance its versatility and enable more comprehensive molecular design, potentially improving its performance in a wider range of applications.

**Questions For Authors:**

See above

**Relation To Broader Scientific Literature:**

Yes

**Theoretical Claims:**

Yes

---

> ### Author Rebuttal · Authors · 2025-04-01
>
> We greatly appreciate Reviewer mgY4 for the review and thoughtful feedback. Below, we provide detailed point-by-point responses to address your remaining concerns.
>
> **[mgY4] asked for additional metrics on text-to-molecule generation.**
>
> As the reviewer suggested, we measured the uniqueness, novelty, and prompt alignment score for the prompts we’ve tested using 1,000 samples. Validity is the proportion of generated SMILES that is valid. Uniqueness is the proportion of valid SMILES that are unique.  The “align” score is the proportion of unique SMILES that match the given prompt. Novelty is the proportion of the unique SMILES that are not included in the training dataset. The alignment score was measured with pattern matching with the substructure described by the prompt.
>
> We observed that even when stochastic sampling was enabled, AR models struggled to generate various samples from a single prompt. LDMol can generate molecules that align better with various hand-written prompts. Furthermore, its outputs were much more diverse than the previous AR models.
>
> | | Models  | Validity(V) | Uniqueness(U) | Align(A) | VxUxA | Novelty |
> |---:|---:|:---:|:---:|:---:|:---:|:---:|
> |Case (a)|molT5|**0.996**|0.006|**1.000**|0.006|N/A|
> | |bioT5+|0.846|0.028|0.625|0.015|N/A|
> | |LDMol|0.910|**0.951**|**1.000**|**0.865**|0.988|
> |Case (b)|molT5|0.927|0.012|0.818|0.009|N/A|
> | |bioT5+|**1.000**|0.573|0.782|0.448|N/A|
> | |LDMol|0.989|**0.960**|**0.906**|**0.860**|0.958|
> |Case (c)|molT5|0.783|0.072|0.643|0.036|N/A|
> | |bioT5+|**1.000**|0.160|**0.750**|0.120|N/A|
> | |LDMol|0.955|**0.861**|0.688|**0.566**|0.780|
> |Case (d)|molT5|0.995|0.002|0.500|0.001|N/A|
> | |bioT5+|**1.000**|0.015|**0.733**|0.011|N/A|
> | |LDMol|0.956|**0.849**|0.703|**0.571**|0.842|
> |Case (e)|molT5|0.956|0.015|0.571|0.008|N/A|
> | |bioT5+|**1.000**|0.035|0.086|0.003|N/A|
> | |LDMol|0.996|**0.187**|**0.595**|**0.111**|0.667|
> ___
>
> **[mgY4] suggested the potential expansion toward different types of data and conditions.**
>
> We appreciate the reviewer’s expectation for the potential applicability of our approach towards the other types of data, and we also regard the expanding of carefully designed latent diffusion models as possible future works. As the reviewer suggested, this can be applicable to both different target chemical data domain itself or various biochemical conditions.

---

### Official Review · Reviewer_qAD8 · 2025-03-10

**Overall Recommendation:** 3

**Summary:**

This paper proposes a text-conditioned molecule (SMILES) generation model based on latent diffusion. LDMol learns a structurally informative latent space through contrastive learning, and surpasses AR models. The authors claim LDMol is the first diffusion model outperforms AR models in text-to-mol generation.

**Claims And Evidence:**

1. LDMol verifies the potential of latent diffusion in molecule generation
2. Through comtrastive learning, LDMol learns a chemically informative latent space, which is important for molecule LDM.

**Essential References Not Discussed:**

N/A

**Experimental Designs Or Analyses:**

1. Structure-aware SMILES latent space with SMILES contrastive learning.
Structure-awareness seems an overclaim. The author can visualize the learned latent representations and see if there are meaning clusters reflecting certain chemical structures.
2. Text-conditioned molecule generation.
LDMol's validity is worse than bioT5, which is not discussed. BLEU score is not a good metric for molecule SMILES generation, and the exact match scores are still low. The conclusions associated with Fig 4 seem overclaimed if there is no further quantitative evidence.
3. Molecule-to-text retrieval is not useful.
The reviewer concerns LDMol can truly align narrative description and chemical structures. Text-to-molecule retrieval seems more helpful in real world applications. Can the authors try text-to-molecule retrieval settings?

**Methods And Evaluation Criteria:**

See experiment review

**Other Comments Or Suggestions:**

No

**Other Strengths And Weaknesses:**

Weakness:
1. Is text-to-SMILES generation really useful? Can you provide some real world cases?
2. See experiment review.

**Questions For Authors:**

Pairwise translation seems easier to hack than retrieval. How to guarantee the model truly aligns the narrative and chemical semantics instead of memorizing some prompts and SMILES tokens? Why not text-2-mol retrieval, which is harder but more useful than text-2-mol generation.

**Relation To Broader Scientific Literature:**

In real world applications, structure- or function-conditioned molecule generation/retrieval is more helpful, especially in drug discovery or de novo design. For example, virtual screening, affinity prediction, drug design.

Gao, Bowen, et al. "Drugclip: Contrastive protein-molecule representation learning for virtual screening." Advances in Neural Information Processing Systems 36 (2023): 44595-44614.
Wang, Renxiao, et al. "The PDBbind database: methodologies and updates." Journal of medicinal chemistry 48.12 (2005): 4111-4119.
Luo S, Guan J, Ma J, et al. A 3D generative model for structure-based drug design[J]. Advances in Neural Information Processing Systems, 2021, 34: 6229-6239.

**Theoretical Claims:**

N/A

---

> ### Author Rebuttal · Authors · 2025-04-01
>
> Thank you for your thoughtful feedback and for considering our work, and we appreciate your time and evaluation. Below we provide point-by-point responses on your questions and concerns.
>
> **[qAD8] asked for visualization of the latent space and its structural information.**
>
> To visualize the structural information encoded in the latent space of our encoder, we prepared 10 molecular clusters that contain 100 molecules, each sharing the common Murcko scaffold. Then, we obtained their latent vector from the LDMol encoder and visualized them in 2D via UMAP[1]. We also plotted the latent vector of 5,000 randomly sampled molecules. As shown in the following Figure 1 [[LINK]](https://docs.google.com/presentation/d/1UV4vMpHT4hrGa1I9PdswLgZ9PJ7vlPH22xbgFKpfo1U/edit#slide=id.p), the molecules with shared Murcko scaffold[2] have formed the clusters on the latent vector space.
> ___
> **[qAD8] expressed a concern on the text-to-molecule generation performance.**
>
> We kindly note that we had discussed that MolXPT and bioT5 had higher validity than ours in the manuscript(line 307\~310, right column), yet they shorted in every metric for the similarity between the ground truth. We insist that the primary role of a text-to-molecule model is to generate a molecule that meets the user prompt, thus, similarity metrics should be prioritized when evaluating models.
>
> We agree that character-wise metrics like BLEU and Levenshtein distance are not appropriate for SMILES generation tasks, and we included them as a convention for many studies in this benchmark. LDMol still showed the major performance improvement from the previous AR models on chemical similarities, including a 6.7~15.2%p increases in fingerprint similarities and the state-of-the-art exact match ratio of 0.530 in ChEBI-20 benchmark.
>
> For the case study in Figure 4, we measured the uniqueness, novelty, and the prompt alignment score using 1,000 samples. The alignment score was measured with pattern matching with the structure described by the prompt. Please refer to the table in the response to the reviewer `mgY4` below. It is shown that LDMol can generate molecules that match with various hand-written prompts, and its output is more diverse than the previous AR models.
>
> ___
> **[qAD8] questioned the usefulness of molecule-to-text retrieval and suggests text-to-molecule retrieval.**
>
> We want to emphasize that the primary contribution of LDMol is in the text-to-molecule generation, and the downstream tasks were done to demonstrate its applicability as a diffusion model compared to AR-based generative models. That said, we performed a paragraph-level text-to-molecule retrieval task on the PCdes test set and the MoMu dataset, measuring 64-way accuracy. Here, we had to estimate the ELBO of each query molecule with given text input using noise prediction error across different noise levels, and the query with the highest likelihood was selected. Although the exact calculation for ELBO would take 1,000 noise predictions per pair, LDMol already showed comparable performance with the baseline representation models with a rough estimation of NFE=25 while being a successful generative model at the same time.
>
> |  | mol-to-text(PCdes) | mol-to-text(MoMu) | text-to-mol(PCdes) | text-to-mol(MoMu) |
> |---:|:---:|:---:|:---:|:---:|
> |SciBERT|62.6|1.38|61.8|1.56|
> |KV-PLM|77.9|1.51|77.0|1.60|
> |MoMu-S|80.6|45.7|80.2|46.0|
> |MoMu-K|81.1|46.2|81.4|45.7|
> |MoleculeSTM|81.4|67.6|78.9|64.1|
> |MolCA|86.4|73.4|**84.8**|72.8|
> |LDMol(n=25)|**90.3**|**87.1**|83.3|**74.0**|
> ___
> **[qAD8] questioned the usefulness of text-to-molecule generation.**
>
> As demonstrated by the applications of LLM on various data domains[3][4], natural text is a modality that enables the incorporation of various conditions like molecular properties, interactions, etc. This makes the generative model utilizing text conditions more applicable and expandable compared to the models trained with specific condition types. While its pratical utility still needs to be improved for real-world usage, text-conditioned molecule generation has received a growing interest as we noted in Related Works, and LDMol achieved the state-of-the-art performance over the baselines.
> ___
>
> [1] UMAP: Uniform Manifold Approximation and Projection for Dimension Reduction, arxiv 2018.
>
> [2] The Properties of Known Drugs. 1. Molecular Frameworks, Journal of Medical Chemistry 1996.
>
> [3] TableLLM: Enabling Tabular Data Manipulation by LLMs in Real Office Usage Scenarios, arxiv 2024.
>
> [4] Can Language Models Solve Graph Problems in Natural Language?, NeurIPS 2023.

---

> > ### Comment · Reviewer_qAD8 · 2025-04-03
> >
> > 1. I recognize the learned representation is structure-aware. Thanks for visulization results.
> > 2. The fingerprint similarity looks good. I agree the model can generate molecules satisfying prompts.
> > 3. The case study in Fig 4 are too simple. Can you try other prompts involving drug-likeness, e.g., QED, SA, and report the mean/median scores accordingly?
> > 4. Thanks for providing text-to-molecule retrieval results. The results look helpful. Can you explain why scores are much lower on MoMu?
> > 5. Can LLM really learn molecule/protein-molecule interactions? Can you provide some more concrete reference to related work? If that's true, LLM will be helpful to real world applications, e.g., drug design, enzyme design.
> >
> > Thanks for the authors' response, many of my concerns are addressed. From my point, SMILES is outdated and less useful compared to graph or 3d representations. I will encourage you to expand your work in structure-based applications.
> >
> > I will raise my score.

---

> > > ### Author Response · Authors · 2025-04-04
> > >
> > > Thank you for raising the score. We are pleased that our rebuttal successfully addressed your previous concerns. Below, we include responses to your additional comments:
> > >
> > > - We found that most of the prepared available molecule-text training data describe the structures and functional groups, thus the prompts in Figure 4 were designed to test LDMol’s ability on various ranges of structural descriptions. In return, the current LDMol often struggles with statements rare in the training data, such as druggability, as we stated in the Conclusion.
> > > That said, we list the QED / SAscore of accordingly generated molecules from LDMol and several baselines below. Although QED still has room for improvement for real-life applicability, LDMol showed better alignment than molT5 and bioT5+. We believe the model performance can be enhanced further with the emergence of richer text-molecule pair data.
> > >
> > > | Prompts | metric | molT5 | bioT5+ | LDMol(ours) |
> > > | :-: | :-: | :-: | :-: | :-: |
> > > | “This molecule is a drug-like molecule.” | QED[↑] | 0.544 | 0.480 | **0.619** |
> > > | “This molecule is synthesizable.” | SAscore[↓] | 2.852 | 3.165 | **2.767** |
> > >
> > > - Although there would be various reasons for this phenomenon, we observed that the MoMu test set contains more difficult text descriptions that require database knowledge rather than the understanding of the statement(e.g. ~15% of the text description has a format of "XX is a natural product found in YY."), which can be one of the causes of why all models’ retrieval performance decreases on the MoMu test set.
> > >
> > > - Despite connecting biochemical information into LLM is beyond the scope of our work, recent works[1] aim to solve desirable prediction[2] or interaction[3]  tasks under the control of natural language by incorporating LLM with molecules or proteins, via explicit text-like data format(e.g. SMILES, AA sequence, etc.)  or appropriate encoders.
> > > ___
> > > [1] Artificial intelligence enabled ChatGPT and large language models in drug target discovery, drug discovery, and development, Molecular Therapy-Nucleic Acids, 2023.
> > >
> > > [2] Can Large Language Models Empower Molecular Property Prediction?, arxiv 2023.
> > >
> > > [3] ProLLM: Protein Chain-of-Thoughts Enhanced LLM for Protein-Protein Interaction Prediction, COLM 2024.

---

### Official Review · Reviewer_bRG5 · 2025-03-12

**Overall Recommendation:** 3

**Summary:**

The paper proposes a latent diffusion model for text-conditioned molecule generation. The authors claim that their primary innovation lies in introducing a contrastive learning approach to capture molecular structural features from SMILES sequences. In tasks such as molecule-to-text retrieval and text-guided molecule editing, this method demonstrates certain improvements compared to autoregressive-based approaches.

## update after rebuttal
I have read all the feedback and raised my score.

**Claims And Evidence:**

Some of the claims in this paper are far-fetched. I have outlined specific comments in the weaknesses section.

**Essential References Not Discussed:**

Several similar text-conditioned generative models based on diffusion frameworks have already been proposed. I have listed the relevant works in the weaknesses part.

**Experimental Designs Or Analyses:**

Some baselines lack comparative analysis. I have detailed specific comments in the weaknesses section.

**Methods And Evaluation Criteria:**

Yes.

**Other Comments Or Suggestions:**

1. The examples in Figure 4 only present the validity of the conditionally generated molecules. Whether these molecules meet the expectations of the given conditions should also be showcased.

2. The description of the text-guided molecule editing task is not sufficiently clear. The authors only refer to the DDS method. A brief outline of the steps should be included in the paper.

3. The model architecture of the encoder should be presented more clearly. At least, an architectural framework should be provided in the appendix.

**Other Strengths And Weaknesses:**

**Strengths:**
1. The authors’ attempt to address text-conditioned molecule generation is a worthwhile and exploratory field.
2. The authors’ use of different SMILES sequences of the same molecule for contrastive learning is a novel approach.

**Weaknesses:**
1. Some claims in the paper are far-fetched:

   a) The authors claim that the SMILES encoder can learn molecule structure information, but the proposed contrastive learning strategy relies more on whether the SMILES sequences originate from the same molecule. Structural differences are not explicitly modeled or learned.

   b) The authors claim, "By preparing an encoder to provide a chemically useful and interpretable feature space, our model can more easily connect the molecule data with the highly complicated condition of natural texts." How does the encoder trained with the proposed contrastive learning strategy achieve easier alignment with the text space? Additionally, the term "interpretable" lacks detailed explanation and validation.

   c) The authors claim they proposed "the first diffusion model that successfully surpassed autoregressive models in textual data generation." Such models are not novel, as many diffusion models have already been proposed for text-guided molecule generation, such as:

      [1] Periodic Materials Generation using Text-Guided Joint Diffusion Model. ICLR 2025.

      [2] Text-Guided Molecule Generation with Diffusion Language Model. AAAI 2024.

      [3] Hierarchical Graph Latent Diffusion Model for Conditional Molecule Generation. CIKM 2024.

      [4] Text-guided small molecule generation via diffusion model. iScience, 2024, 27(11).

2. The experimental results are insufficient:

   a) The authors only compared autoregressive models. Text-guided diffusion models should also be included as baselines.

   b) An ablation study on the contrastive learning component is necessary.

**Questions For Authors:**

1. How to understand the statement "we suggest a novel contrastive encoder learning strategy by minimizing mutual information between positive SMILES pairs"?

**Relation To Broader Scientific Literature:**

The key contribution of this paper stems from the latent diffusion framework in the field of image generation.

**Theoretical Claims:**

This paper has no theoretical proofs.

---

> ### Author Rebuttal · Authors · 2025-04-01
>
> We appreciate the reviewer for your detailed comments and valuable suggestions. Below, we provide thorough point-by-point responses to address your concerns.
>
> **[bRG5] asked how the proposed contrastive learning can learn the structural differences.**
>
> We would like to remind the reviewer that in order to minimize the proposed contrastive loss, the feature cosine similarity from the different molecules in the batch needs to be minimized, which enables the encoder to reflect structural differences into their latent output. Please refer to the UMAP[1] visualization of LDMol’s latent space in the following Figure 1 [[LINK]](https://docs.google.com/presentation/d/1UV4vMpHT4hrGa1I9PdswLgZ9PJ7vlPH22xbgFKpfo1U/edit#slide=id.p), where the molecules with shared Murcko molecular scaffold[2] locate closer in the latent space.
> ___
> **[bRG5] asked how our latent space construction helps the alignment with the text conditions.**
>
> Compared to the raw SMILES tokens or structure-unaware latent from $\beta$-VAE(Figure 3-(b)), our encoder provides a latent where the proximity is more structurally meaningful, hence more “interpretable” for the later diffusion model that needs to link the text information. Indeed, fingerprint similarities and FCD on the ChEBI-20 dataset were improved by 7~15%p and 40% from the previous baselines, meaning that LDMol made a reasonable guess even when it was incorrect.
> ___
> **[bRG5] expressed concern about the claim on the model performance.**
>
> While there were many “text-conditioned” chemical diffusion models, we clarify that the term “textual data” generation refers to the text-like generation target of SMILES; most successful chemical diffusion models had targeted graph or point-cloud data, and diffusion models that generate SMILES[3] showed suboptimal performance compared to AR models. We insist that we’ve shown the potential of improving diffusion model performances on text-like data with carefully designed latent space.
> ___
> **Text-guided diffusion models as baselines.**
>
> Please note that TGM-DLM[4] is included as a text-guided molecule diffusion model baseline (line 340). TGM-DLM trained the diffusion model by naively treating the token index as a continuous real value, which led to a severe performance drop compared to LDMol.
> ___
> **Ablation study on contrastive learning.**
>
>  Please refer to the revised ablation studies that we included in the response to the reviewer `YcAV` above. To replace our suggested latent space construction, we tested two common strategies of naive reconstruction and KL-regularized autoencoder. The latent space without any regularization was unable to learn by the diffusion model, and the latent space from $\beta$-VAE had suboptimal performance with notably low validity.
> ___
> **[bRG5] asked the prompt alignment score in the case studies.**
>
> Please refer to the table in the response to the reviewer `mgY4` below, where we measured the prompt alignment score for the prompts we’ve tested using 1,000 samples. The result demonstrates that LDMol showed better and more consistent alignment along various hand-written prompts compared to the AR baselines.
> ___
> **[bRG5] asked for a clearer description of the molecule editing process.**
>
> Our text-guided molecule editing was done similarly to Delta Denoising Score(DDS), where the input data is optimized to match the target prompt by minimizing the difference between the model-predited noise with (source_prompt, source_data) pair and (target_prompt, target_data) pair. Please note that Supplementary material A.3 contains a detailed description and pseudocode of our DDS-based test-guided molecule editing.
> ___
> **The architecture of the encoder.**
>
> The encoder we’ve used consists of 12 bidirectional transformer layers of BERT$_{base}$, with a feature size of 1024 and 16 attention heads. We thank the reviewer for the suggestion, and we’ll include this information in Supplementary material A.2.
> ___
> **[bRG5] questions the statement of minimizing mutual information.**
>
> The mentioned statement describes our contrastive latent space construction with SMILES enumeration, compared to previously suggested molecular contrastive learning methods[4].
> Since SMILES enumeration provides all possible variations under the SMILES grammar, the enumerated SMILES pair has minimal mutual information(i.e., the connectivity of atoms and bonds) compared to simple or local augmentations. As a result, while most contrastive learning focuses on “extracting” certain desired features, our contrastive learning can “fully preserve” the structural information as the augmentation invariant to become an autoencoder.
> ___
>
> [1] UMAP: Uniform Manifold Approximation and Projection for Dimension Reduction, arxiv 2018.
>
> [2] The Properties of Known Drugs. 1. Molecular Frameworks, Journal of Medical Chemistry 1996.
>
> [3] Text-Guided Molecule Generation with Diffusion Language Model, AAAI 2024.
>
> [4] Graph contrastive learning with augmentations. NeurIPS 2020.

---

> > ### Comment · Reviewer_bRG5 · 2025-04-09
> >
> > Thank you to the authors for their responses. I appreciate the additional experiments on prompt alignment scores for text-guided conditional generation as well as the ablation study on contrastive learning. These efforts help to strengthen the empirical support of the work, and accordingly, I will raise my score.
> >
> > That said, I still find that some of the claims in the paper may be overstated and would encourage the authors to revise the corresponding language for greater precision and clarity. Specifically:
> >
> > a) The statement “a latent where the proximity is more structurally meaningful, hence more interpretable for linking the text information” lacks a well-substantiated connection between structural properties and text semantics. It remains unclear how the notion of "structural meaningfulness" in the latent space translates into improved interpretability with respect to textual descriptions. Additional analysis or justification would be helpful to substantiate this claim.
> >
> > b) While the additional clustering results are appreciated, they do not convincingly support the assertion that “the proposed method learns structural information.” In the absence of explicit modeling of structural features, it is difficult to determine what aspect of the data the model is actually leveraging. It remains a plausible alternative that the model is simply grouping molecules based on elemental composition rather than higher-level structural similarity. To more convincingly support the claim of structural learning, it would be helpful to demonstrate that the model can cluster molecules that are structurally similar despite differing in atomic composition.
> >
> > I encourage the authors to refine these claims in the final version to more accurately reflect what is supported by the current evidence.

---

> > > ### Author Response · Authors · 2025-04-09
> > >
> > > Thank you for raising the score towrads acceptance, and we are pleased that our rebuttal successfully addressed your previous concerns. Due to the time constraint, we include responses to your additional comments below, and we assure that additional analyses with more refined statements will be included in the final draft.
> > >
> > > - Assuming most of the controllable conditions(e.g. functional groups, internal properties, etc.) has unavoidable correlation with the molecule structure, we insist that data domain retaining structural information would benefit the generative model, regardless the modality of the condition(e.g. natural text). For instance, training conditional diffusion model would be easier if the molecules satisfying the condition forms certain manifold or clusters.
> > >
> > > - We assure that the group of molecules clustered in UMAP latent space only shares a scaffold as a high-level molecular structure, and the atomic-level composition of each molecule varies by their side chains and functional groups. From the clustering of the molecules with similar overall molecule structures, we concluded that the LDMol latent space retains structural informations compared to naive reconstruction-based latents.
> > >
> > > We thank again the reviewer for the constructive feedback and raising the score.

---

### Official Review · Reviewer_YcAV · 2025-03-14

**Overall Recommendation:** 3

**Summary:**

- This paper proposes a SMILE-based latent diffusion method for text-driven molecule generation task.
This paper augments the data by aligning enumerated SMILES with the traditional contrastive learning approach at the pretraining stage to ask the model to learn the invariant features from SMILES.
With the pretrained molecule encoder, a decoder that can successfully reconstruct the molecule, and a frozen text encoder, this paper trains a latent diffusion model that shows competitive performance with other Auto-regressive SMILE-based models.

**Claims And Evidence:**

- The assumption of previous methods treated enumerated SMILE differently is well proved in Figure 3 (b) which shows the distance gets diminished by contrastive learning.
- The statistic of enumerated SMILEs needs to be discussed, need some introduction on the original datasets and the augmented datasets. For instance, what are the sizes of molecules? What are the elements and function groups of molecules? What are the lengths and the informative text descriptions? How are they aligned and how many enumerated SMILE pairs on average?
- The assumption that contrastive learning would help the model to generate better molecules should be further discussed and needs an experiment on the final downstream tasks. Even if Figure 3 (b) shows the distance between enumerated SMILES was large without contrastive learning, it may still need to show the performance under original settings. If the main contribution is introducing contrastive pre-training, then authors may need to try the proposed method on those Auto-regressive models for showing better performance. It is hard to conclude whether the competitive result is from pre-training, compression layers, or diffusion models.

**Essential References Not Discussed:**

- As the one of the early latent diffusion methods in text-driven molecule generation, the related references are included.

**Experimental Designs Or Analyses:**

I have checked the experiment design. Related concerns:
The paper should include a comprehensive evaluation of the model's final performance on downstream tasks within the experimental section. Simply claiming that contrastive learning benefits the final-stage performance without providing empirical evidence is insufficient. A quantitative comparison with baseline methods is necessary to demonstrate the advantages of the proposed approach.
- The current ablation study lacks clarity and depth. While the compression layer is not presented as the primary contribution of the paper, it evidently plays a crucial role in downstream task performance. The authors should conduct a more thorough investigation into the function of this layer by analyzing the latent space representations. Providing insights into how the compression layer influences model performance would significantly improve the final performance.
- The dataset augmentation should be explicitly examined in the experiments. Since longer SMILES strings naturally generate more enumerated pairs, is it necessary to use all of them? If not, what is the optimal amount of augmented data required for effective training. Specifically, the authors should investigate how varying the amounts of augmented SMILES pairs to abstract away from explicit SMILES grammar constraints, or is there still be grammar things left in the model?

**Methods And Evaluation Criteria:**

Yes, authors follow the previous works such as MolT5. They compared the performance among Auto-regressive models under same metrics.

**Other Comments Or Suggestions:**

N/A

**Other Strengths And Weaknesses:**

__Strengths:__
- Clearly written, easy to follow, and understand the paper, the figures are very clear and helpful for understanding.
- Originality: the thought of data augmentation and contrastive learning are simple and implemented in the unconditional molecule generation before, but the originality of this paper is enough for text-driven molecule generation at this early stage.

__Weakness:__ please check the __Experimental Designs Or Analyses__ and __Claims And Evidence__.

**Questions For Authors:**

Please check the __Experimental Designs Or Analyses__ and __Claims And Evidence__.

**Relation To Broader Scientific Literature:**

- The major contribution of this paper is proposing a pretraining stage with augmented dataset and contrastive learning to help model learn in-variant features from SMILE.
- The second contribution is as one of the early latent diffusion methods in text-driven molecule generation, it shows comparable performance to AR-based methods.

**Theoretical Claims:**

Yes, I have checked the theoretical part. The major theoretical claim is that contrastive learning will help the model learn invariant features from SMILES, and it is well stated by Figure 3.

---

> ### Author Rebuttal · Authors · 2025-04-01
>
> We thank Reviewer YcAV for the constructive feedback. Below, we provide point-by-point responses on your questions and concerns.
>
> **[YcAV] asked for the statistics of the enumerated SMILES.**
>
> Please note that the SMILES enumeration we’ve introduced is done by different SMILES constructions of the same molecule, and does not modify the given molecule. Therefore, the molecule size, atoms, bonds, etc., are the same as in the initial dataset PubChem, which contains a wide range of general molecules. We used 10 million randomly sampled molecules from PubChem to construct contrastive latent space, where the SMILES length over 500 were removed. No paired text descriptions were used in this training phase.
> ___
>
> **[YcAV] requested the analysis and ablation study on the effect of contrastive learning on the generation task.**
>
> Please refer to the revised ablation studies below. The first two rows (a) and (b) show the model performance by replacing contrastive learning-based latent space into two common latent space construction strategies of naive reconstruction and KL-regularized autoencoder. The latent space without any regularization was unable to learn by the diffusion model, and the latent space from beta-VAE had suboptimal performance with notably low validity.
>
> | Models  | Latent space construction | stereoisomer hard-negatives | laten space compression | AE recon. acc.[↑] | ChEBI-20 Validity[↑] | ChEBI-20 Match[↑] | ChEBI-20 FCD[↓] |
> |---:|:---:|:---:|:---:|:---:|:---:|:---:|:---:|
> | LDMol | Contrastive loss | O | Linear | 0.983 | **0.941** | **0.530** | **0.20** |
> | (a) | None | (N/A) |  | **1.000** | 0.019 | 0.000 | 58.6 |
> | (b) | KL regularization | (N/A) |  | 0.999 | 0.847 | 0.492 | 0.34 |
> | (c) |  | X |  | 0.891 | 0.939 | 0.278 | 0.24 |
> | (d) |  |  | None | 0.964 | 0.022 | 0.000 | 67.9 |
> | (e) |  |  | transformer layers | 0.986 | 0.565 | 0.084 | 2.19 |
>
> Although we appreciate the reviewer’s suggestion, our latent construction of contrastive learning is currently inapplicable to AR models since they do not utilize any latent domain. It might be possible to apply contrastive loss into intermediate feature space similar to [1], but we believe this is beyond the scope of our work.
> ___
>
> **[YcAV] asked for further analysis on the compression layer**
>
> Since the latent space compression was done by a single linear layer, its role is to reduce the dimension for the diffusion modeling rather than the extra curation of the features. Please refer to the UMAP[1] latent visualization in the following Figure 1 [[LINK]](https://docs.google.com/presentation/d/1UV4vMpHT4hrGa1I9PdswLgZ9PJ7vlPH22xbgFKpfo1U/edit#slide=id.p), where the feature space is structurally informative before and after the compression layer.
>
> Also, we consistently observed that the role of the compression layer should be minimized, as we had to leverage the smooth and regulated latent space from the contrastive learning. Please note that the model performance degraded as the diffusion model’s target domain deviates from the contrastive latent space, even when the compression layer capacity is increased(see row (e) of the table above).
> ___
>
> **[YcAV] suggested further clarification and examination of the SMILES enumeration.**
>
> Longer SMILES have more enumerated SMILES pairs as the reviewer noted, and we randomly selected one possible enumeration for each pre-training epoch. Therefore, the number of enumeration pairs for each training data that the model encountered are mostly equal, except for the molecules that are extremely small. Even fixing the single enumeration for each training data was enough for the encoder to learn the enumeration-invariant features, as we show the histogram of enumerated SMILES pair distances in the following Figure 2 [[LINK]](https://docs.google.com/presentation/d/1UV4vMpHT4hrGa1I9PdswLgZ9PJ7vlPH22xbgFKpfo1U/edit#slide=id.g3473ce19713_0_1).
>
> ___
>
> [1] UMAP: Uniform Manifold Approximation and Projection for Dimension Reduction, arxiv 2018.
>
> [2] Representation Alignment for Generation: Training Diffusion Transformers Is Easier Than You Think, ICLR 2025

---

### Decision · Program_Chairs · 2025-05-01

**Decision:**

Accept (poster)

**Comment:**

This paper uses existing SMILES enumerations to improve a representation via contrastive learning that's to be used in a latent diffusion to generate molecules conditioned on text. While a straightforward assembly of techniques already in the literature, the results look strong enough for this paper to be accepted. The main weakness was the writing was often vague and seemed like it overclaimed as noted by several of the reviewers. The authors should address this writing issue and work in some of the additional results and ablations in the author response in the final version of the paper.